# Global, regional, and national estimates of the impact of a maternal *Klebsiella pneumoniae* vaccine: A Bayesian modeling analysis

**Chirag K. Kumar** [1], **Kirsty Sands** [2], **Timothy R. Walsh** [2], **Seamus O'Brien** [3],
**Mike Sharland** [4], **Joseph A. Lewnard** [5], **Hao Hu** [6], **Padmini Srikantiah** [6],
**Ramanan Laxminarayan** [1,7] *

**1** Princeton University, Princeton, New Jersey, United States of America, **2** Ineos Oxford Institute for Antimicrobial Resistance, Department of Zoology, Oxford, United Kingdom, **3** Global Antibiotic Research and Development Partnership, Geneva, Switzerland, **4** Center for Neonatal and Paediatric Infection (CNPI), Institute of Infection and Immunity, St George's University of London, London, United Kingdom, **5** Division of Epidemiology, School of Public Health, University of California at Berkeley, Berkeley, California, United States of America, **6** Bill & Melinda Gates Foundation, Seattle, Washington, United States of America, **7** One Health Trust, Bengaluru, India

* ramanan@onehealthtrust.org

**Data Availability Statement:** All underlying model code and freely and publicly available data (i.e., CHERG and GBD data, maternal tetanus vaccination rates) that were used in this analysis are available from https://github.com/ChiragKumar9/KpVaccine. The CHAMPS data are

## Abstract

### Background

Despite significant global progress in reducing neonatal mortality, bacterial sepsis remains a major cause of neonatal deaths. *Klebsiella pneumoniae* (*K. pneumoniae*) is the leading pathogen globally underlying cases of neonatal sepsis and is frequently resistant to antibiotic treatment regimens recommended by the World Health Organization (WHO), including first-line therapy with ampicillin and gentamicin, second-line therapy with amikacin and ceftazidime, and meropenem. Maternal vaccination to prevent neonatal infection could reduce the burden of *K. pneumoniae* neonatal sepsis in low- and middle-income countries (LMICs), but the potential impact of vaccination remains poorly quantified. We estimated the potential impact of such vaccination on cases and deaths of *K. pneumoniae* neonatal sepsis and project the global effects of routine immunization of pregnant women with the *K. pneumoniae* vaccine as antimicrobial resistance (AMR) increases.

### Methods and findings

We developed a Bayesian mixture-modeling framework to estimate the effects of a hypothetical *K. pneumoniae* maternal vaccine with 70% efficacy administered with coverage equivalent to that of the maternal tetanus vaccine on neonatal sepsis infections and mortality. To parameterize our model, we used data from 3 global studies of neonatal sepsis and/ or mortality—with 2,330 neonates who died with sepsis surveilled from 2016 to 2020 undertaken in 18 mainly LMICs across all WHO regions (Ethiopia, Kenya, Mali, Mozambique, Nigeria, Rwanda, Sierra Leone, South Africa, Uganda, Brazil, Italy, Greece, Pakistan, Bangladesh, India, Thailand, China, and Vietnam). Within these studies, 26.95% of fatal neonatal sepsis cases were culture-positive for *K. pneumoniae*. We analyzed 9,070 *K.*

freely available online upon request. In this analysis, we used CHAMPS Level 2: De-Identified Data, which are all available at the following link after signing a data transfer agreement: https://champshealth.org/data/. BARNARDS data are owned by the Ineos Oxford Institute for Antimicrobial Resistance. They are freely available upon request from the principal investigators of the study. For details, please reach out to Kathryn Thomson at kathryn.thomson@zoo.ox.ac.uk. NeoObs data are owned by the Global Antibiotic Research and Development Partnership and are freely available upon request. For details, please contact Sally Ellis at sellis@gardp.org. The genomes analyzed in this paper are all available through the European Nucleotide Archive under project number PRJEB33565. Individual accession numbers for all genomes analyzed are also available as part of S1 Table.

**Funding:** This work was supported by the Bill & Melinda Gates Foundation (grant OPP1190803; to RL), the Smith-Newton Environmental Fellowship provided through the High Meadows Environmental Institute at Princeton University (no grant number; to CKK), the US Centers for Disease Control and Prevention (grant 21IPA2113462; to RL), and the US National Science Foundation (grant CCF1918628; to RL). The funders had no role in study design, data collection and analysis, decision to publish, or preparation of the manuscript.

**Competing interests:** We have read the journal's policy and the authors of this manuscript have the following competing interests: two study authors (PS and HH) are employed by an organization that funded the study. They provided input in the writing of the report, acting in a purely scientific capacity. In particular, PS provided critical expertise on *K. pneumoniae* epidemiology and vaccine development. HH provided valuable insights into modeling bacterial pathogens.

**Abbreviations:** AGRG, aminoglycoside-resistant gene; AMR, antimicrobial resistance; ARG, antimicrobial resistance gene; BARNARDS, Burden of Antibiotic Resistance in Neonates from Developing Societies; CFR, case fatality ratio; CHAMPS, Child Health and Mortality Prevention Surveillance; CHERG, Child Health Epidemiology Reference Group; CRG, carbapenem-resistant gene; CrI, credible interval; CSF, cerebrospinal fluid; GBD, Global Burden of Disease; LHS, Latin hypercube sampling; LMIC, low- and middle-income country; MDR, multidrug resistance; MITS, minimally invasive tissue sampling; WHO, World Health Organization.

*pneumoniae* genomes from human isolates gathered globally from 2001 to 2020 to quantify the temporal rate of acquisition of AMR genes in *K. pneumoniae* isolates to predict the future number of drug-resistant cases and deaths that could be averted by vaccination.

Resistance rates to carbapenems are increasing most rapidly and 22.43% [95th percentile Bayesian credible interval (CrI): 5.24 to 41.42] of neonatal sepsis deaths are caused by meropenem-resistant *K. pneumoniae*. Globally, we estimate that maternal vaccination could avert 80,258 [CrI: 18,084 to 189,040] neonatal deaths and 399,015 [CrI: 334,523 to 485,442] neonatal sepsis cases yearly worldwide, accounting for more than 3.40% [CrI: 0.75 to 8.01] of all neonatal deaths. The largest relative benefits are in Africa (Sierra Leone, Mali, Niger) and South-East Asia (Bangladesh) where vaccination could avert over 6% of all neonatal deaths. Nevertheless, our modeling only considers country-level trends in *K. pneumoniae* neonatal sepsis deaths and is unable to consider within-country variability in bacterial prevalence that may impact the projected burden of sepsis.

## Conclusions

A *K. pneumoniae* maternal vaccine could have widespread, sustained global benefits as AMR in *K. pneumoniae* continues to increase.

## Author summary

### Why was this study done?

- Approximately 1 million newborns yearly die within the first 4 weeks of life due to bacteria infecting their bloodstream with *Klebsiella pneumoniae* (*K. pneumoniae*) as the leading cause of such infections.

- There have been numerous recent advancements in developing viable *K. pneumoniae* vaccine and antibody-based treatments in preclinical models, with some treatments reaching Phase 1 clinical trials.

- The impacts of vaccination must be quantified and may prove useful to prioritizing vaccine distribution and better understanding the burden of sepsis.

### What did the researchers do and find?

- Using a Bayesian mixture-model based on clinical surveillance of neonatal sepsis, we present country-specific estimates for the number of deaths and cases of antimicrobial-resistant neonatal sepsis caused by *K. pneumoniae* that would be averted if a vaccine with 70% efficacy were given to pregnant mothers.

- We find that most cases of *K. pneumoniae* neonatal sepsis are resistant to first-line treatments, such as ampicillin and gentamicin.

- We estimate that a vaccine with 70% efficacy could prevent 399,015 [95th percentile credible interval (CrI): 334,523 to 485,442] cases and 80,258 [CrI: 18,084 to 189,040] neonatal deaths.

### What do these findings mean?

- A maternal vaccine that confers newborns with protection from *K. pneumoniae* infection could reduce neonatal sepsis deaths in many low- and middle-income countries (LMICs) by nearly 15%.

- This would help to achieve targets set by the World Health Organization (WHO) for improved child health globally and to mitigate inequities in neonatal survival in low- and middle-income settings compared to high-income settings.

- Reducing cases of neonatal sepsis by vaccination could also contribute to reduced antibiotic use, subsequent improvements in antimicrobial resistance (AMR) rates, and a reduction in healthcare utilization and expenditure.

## Introduction

Each year, an estimated 800,000 newborns die within the first 4 weeks of life due to sepsis [1–3]. Several studies, including most recently the Child Health and Mortality Prevention Surveillance (CHAMPS) [4] and the Burden of Antibiotic Resistance in Neonates from Developing Societies (BARNARDS) [5], indicate that *Klebsiella pneumoniae* is the leading cause of neonatal sepsis (Fig A in S1 Text). Increasing multidrug resistance (MDR) and lack of access to appropriate antibiotics contribute to high levels of morbidity and mortality among *K. pneumoniae* sepsis cases [6–8]. Though improved sanitation and infection control could limit *K. pneumoniae* transmission, they can be challenging to implement in many resource-poor settings [9]. A maternal vaccine against *K. pneumoniae* that confers protection to newborns via transplacental antibody transfer could reduce the burden of neonatal sepsis, but the extent of corresponding reductions in sepsis remains unknown.

Primary approaches to developing a *K. pneumoniae* vaccine have targeted the K capsular polysaccharide antigens or the O lipopolysaccharide antigens—both of which demonstrate significant diversity across *K. pneumoniae* strains—or an inactivated whole cell [10]. Animal data from an early polysaccharide vaccine construct demonstrated a robust immune response and subsequent neutralization of *K. pneumoniae* bacterial isolates in mouse models when administered intravenously [11]. A 24-valent *Klebsiella* capsular polysaccharide vaccine tested in humans demonstrated high anti-polysaccharide immunoglobin titers [12], but a clinical trial to validate this vaccine was inconclusive because of a limited supply of materials [13]. More recently, there is growing interest in conjugate polysaccharide vaccines targeting the K and/or O antigens [14–17] with the first such vaccine entering a clinical phase [18,19]. Combined, these data suggest that a maternal vaccine to prevent *K. pneumoniae* demands further attention and investigation. As with previous [20] and future [21] proposed maternal vaccination programs, this vaccine would be administered to pregnant mothers, triggering an immune response that protects the mother and, critically, transfers maternal antibodies to the developing baby, providing protection to the neonate (or young infant) through the period of greatest risk [22]. Vaccination could reduce the overall burden of sepsis and decrease antibiotic use (fewer *K. pneumoniae* sepsis cases) [23] and *K. pneumoniae* nosocomial outbreaks (vaccination generates community-level immunity) [24], which are common in LMICs [5]. Additionally, by averting infection and reducing the need for treatment, vaccination may prevent lengthy

hospital stays for newborn sepsis, thereby reducing the economic burden on LMIC families where healthcare costs are often deferred to the patient [7].

In this paper, we estimated the global, regional, and national impact of a potential maternal vaccine against *K. pneumoniae* deployed at the current levels of coverage of the maternal tetanus vaccine. We also examined the antimicrobial resistance (AMR) of sepsis-causing isolates. Finally, we estimated the continued benefits of such a vaccine in 2030 under future scenarios of increasing AMR.

## Methods

We used data from 3 global clinical data sets of neonatal sepsis and a Bayesian modeling framework to estimate the annual neonatal infections and deaths by country caused by *K. pneumoniae* (although the methodology is applicable to any other sepsis-causing bacterial etiology; Figs O to U in S1 Text). We used target vaccine efficacy and coverage to project the total number of infections and deaths that could be averted using a maternal vaccine. We complemented these analyses with a mathematical model that estimated the fraction of infections and deaths that are resistant to the antibiotics recommended by World Health Organization (WHO) treatment guidelines for neonatal sepsis (ampicillin and gentamicin as a first-line treatment and amikacin and ceftazidime as a second-line treatment) [25] and those most used to treat neonatal sepsis (adding meropenem to the above list) [7].

### Data

Three data sets were used to quantify *K. pneumoniae* neonatal sepsis burden: the Child Health and Mortality Prevention Surveillance (CHAMPS) [4], the Burden of Antibiotic Resistance in Neonates from Developing Societies (BARNARDS) [5], and the Global Neonatal Sepsis Observational Study (NeoObs) [26]. CHAMPS, initiated in 2016, conducts minimally invasive tissue sampling (MITS) on deceased children under 5 to determine causal drivers of death at 8 study sites in 7 low- and middle-income countries (LMICs) across Africa and South-East Asia (Bangladesh, 2 sites in Ethiopia, Kenya, Mali, Mozambique, Sierra Leone, and South Africa). In contrast, BARNARDS and NeoObs are birth-cohort studies that tracked neonates prospectively through their first month of life, evaluating incidence of sepsis, the antibiotic susceptibility of associated isolates, and resulting mortality. BARNARDS surveilled neonates across 12 study sites in 7 countries in Africa and South-East Asia (2 sites in Bangladesh, India, 2 in Pakistan, Ethiopia, 3 in Nigeria, 2 in Rwanda, and South Africa) from 2015 to 2017. Finally, NeoObs reported neonatal sepsis cases, deaths, and antibiotic susceptibility of the isolated bacteria in 19 sites across 11 countries in Europe (Italy, Greece), Latin America (2 sites in Brazil), Africa (Kenya, Uganda, 3 sites in South Africa), South-East Asia (Bangladesh, 3 sites in India, 2 sites in Thailand), and the Western Pacific (3 sites in China, Vietnam) from 2018 to 2020. Table A in S1 Text provides the number of neonates surveilled over the study period by country. We restricted all analyses to cases within each study with culture-confirmed *K. pneumoniae* sepsis in neonates based on isolates obtained from blood or cerebrospinal fluids (CSF) (Fig B in S1 Text).

### Modeling techniques

We used a Bayesian mixture-modeling approach to aggregate data across the 3 studies and produce consistent and robust estimates of neonatal sepsis infections, deaths, and drug resistance distributions. We estimated $p_{s,l}$, the probability that *K. pneumoniae* was the cause of death for a neonate who died from neonatal sepsis, in each location, $l$, and for each study, $s$. We assumed a noninformative uniform prior on $p_{s,l}$ at all locations. We modeled whether

*K. pneumoniae* was present in a clinical sample from a neonate who died of culture-confirmed sepsis as a binomial random variable and analytically solved for the posterior distribution of $p_{s,l}$ (Section 1 in S2 Text):

$$p_{s,l} \sim \beta(N_{kp}^{s,l} + 1, N_d^{s,l} - N_{kp}^{s,l} + 1),$$

where $N_{kp}^{s,l}$ is the number of neonates who were culture-positive for *K. pneumoniae* and died, $N_d^{s,l}$ is the number of neonates who had culture-confirmed sepsis (from any etiology) and died, both quantities at each location, $l$, in each study, $s$. $\beta$ indicates the standard univariate beta distribution.

In countries where multiple studies conducted surveillance (ex., both CHAMPS and BARNARDS had study sites in the country), we obtained multiple independent estimates of the distribution of $p_{s,l}$, one for each study that had a site in the country. We used a mixture-modeling approach to aggregate these distributions so that each independent distribution for $p_{s,l}$ was weighted proportional to the number of neonates sampled in the study at that location:

$$p_l = \sum_{s=1}^{3} \frac{N_d^{s,l}}{N_d^{1,l} + N_d^{2,l} + N_d^{3,l}} p_{s,l}.$$

We used a mixture-modeling approach rather than aggregating the samples into 1 beta distribution to account for subpopulation variability across sites within the same country from different studies. To efficiently characterize the resulting distribution, we drew 10,000 samples using Latin hypercube sampling (LHS) [27]. We chose LHS over another distribution sampling technique such as Monte Carlo because LHS converges to the true distribution faster.

We extended this analysis to estimate the absolute number of deaths attributable to *K. pneumoniae* using data from the United Nations Child Health Epidemiology Reference Group (CHERG) [1] and the University of Washington's Institute for Health Metrics and Evaluation Global Burden of Disease (GBD) [28,29] on the total number of neonatal deaths and the total number of neonatal sepsis deaths. Estimates of $p_l$ were mapped into estimates of the number of deaths and the percentage of all neonatal deaths attributable to *K. pneumoniae* by scaling the estimates of $p_l$ by the estimates with uncertainty from CHERG and GBD for the total number of neonatal sepsis deaths. Unless otherwise mentioned, we report values derived using CHERG data.

We determined the number of vaccine-avertable deaths associated with the deployment of a maternal *K. pneumoniae* vaccine at an effective coverage level equal to that of the maternal tetanus vaccine (median: 90%; range: 38.5% to 100% of pregnant women immunized) [30]. For modeling purposes, we assumed that the efficacy of the vaccine would be 70%: This estimate is based on a conjugate vaccine candidate that targets the 15 most common *K. pneumoniae* capsular serotypes that cause invasive infections in neonates. Current sero-epidemiology suggests that these serotypes account for approximately 70% of neonatal sepsis cases [14]. To determine the avertable cases from avertable deaths, we estimated the case fatality ratio (CFR) for *K. pneumoniae* sepsis using data from BARNARDS. We used the same Bayesian beta framework with a uniform prior described above (we treated whether the neonate died from *K. pneumoniae* or recovered as the binomial event for which we seek to estimate the probability parameter) to estimate the CFR.

To correctly propagate uncertainty across combination products (i.e., when trying to determine the number of cases from the number of deaths using the CFR), we individually multiplied independent samples from the distributions for $p_{s,l}$ and the CFR. We summarized the resulting distribution using the median and 95% credible intervals (CrIs). We use this approach throughout for propagating uncertainty.

Finally, we estimated antibiotic resistance profiles using data from BARNARDS and NeoObs at the WHO regional level (Section 2 in S2 Text). We used the Bayesian beta and mixture-modeling framework with a noninformative prior described earlier to estimate the probability that an isolate was resistant to a given antibiotic.

We extrapolated our estimates from the selected countries for which we had neonatal sepsis surveillance data global estimates for all countries of avertable neonatal sepsis deaths and cases by fitting regression equations. We related the total number of neonatal deaths due to sepsis (for which estimates are available from CHERG) to the estimated number of avertable neonatal deaths if a *K. pneumoniae* vaccine were distributed as described above using standard ordinary least squares. We also calculated the number of avertable antibiotic-resistant neonatal deaths by country using our estimates for antibiotic resistance at the WHO regional level.

To estimate the future benefits of such a vaccine, we analyzed the temporal trends in the presence of AMR genes in *K. pneumoniae* by using isolate genomes available on Pathogen-Watch [31] and in the BARNARDS data set. We limited our analysis to human isolates from clinical sources that were not taken as rectal or carriage samples (S1 Table and Fig C in S1 Text). We gathered metadata on the isolates, including their date, source, and country. In total, we analyzed 9,070 *K. pneumoniae* genomes from 68 countries from 2001 to 2020 (Figs D and E in S1 Text). To identify antimicrobial resistance genes (ARGs) in each genome, we used ABRicate (v1.1.0) with the reference database ResFinder (last updated on June 18, 2022). We fit linear probability model equations to the temporal trends in the number of aminoglycoside-resistant genes (AGRGs) and carbapenem-resistant genes (CRGs) and extrapolated those trends to 2030 (i.e., to predict the future number of AGRGs and CRGs). We used the fit linear probability model to determine the projected increase in ARGs and resistance by 2030 and used that to estimate the increase in the absolute number of aminoglycoside- and carbapenem-resistant neonatal sepsis cases and deaths.

Fig F in S1 Text shows a schematic diagram of the model, and Table B in S1 Text denotes all model parameters.

## Results

### Effect of a *K. pneumoniae* maternal vaccine on neonatal deaths

We initially focused on the 18 countries for which we had surveillance data on bacterial neonatal sepsis (Fig A in S1 Text). A potential vaccine could reduce neonatal sepsis and improve newborn survival (Fig 1). Our estimates of the proportion of neonatal deaths averted with maternal *K. pneumoniae* vaccination are consistent whether we use CHERG or GBD data as a reference source. Under the assumption that *K. pneumoniae* vaccine coverage matches current effective maternal tetanus vaccination coverage levels [30], a median estimated 57,392 [95th percentile Bayesian CrI: 22,248 to 125,803] deaths, accounting for 3.83% [CrI: 1.48 to 8.40] of newborn fatalities, could be averted each year in the initial 18 countries analyzed. We find a similar benefit in reducing neonatal sepsis cases, with an estimated 286,499 [CrI: 239,259 to 346,496] cases averted yearly.

The benefits of a maternal *K. pneumoniae* vaccine vary across countries. While the relative impact of vaccination on neonatal mortality ranges a few percentage points (Fig 1A), the estimates for the reduction in total neonatal sepsis deaths and cases span orders of magnitude across countries, reflecting the variability in the burden of neonatal deaths across countries. We project that vaccination will have the greatest relative effect on neonatal mortality in Ethiopia, Rwanda, and Sierra Leone and a smaller effect in Greece, China, and Pakistan (Fig 1A), countries where a lower fraction of neonatal sepsis deaths were associated with *K. pneumoniae* (Fig A in S1 Text). However, although the relative impact of vaccination on neonatal mortality

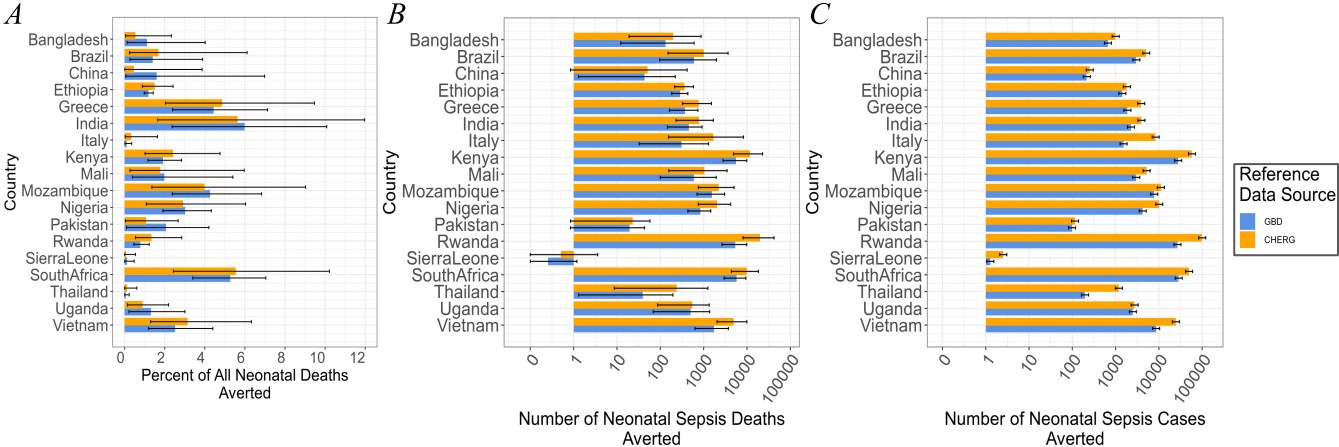

**Fig 1. A maternal vaccine reduces global *K. pneumoniae* burden.** (A) Median estimated percent of neonatal deaths that are avertable with a maternal vaccine with respect to all neonatal deaths in a given country. (B) Median estimated number of neonatal deaths that are avertable with a maternal vaccine in a given country. (C) Median estimated number of neonatal cases of sepsis that are avertable with a maternal vaccine in a given country. GBD indicates data from the Global Burden of Disease study and CHERG indicates data from the United Nations Child Health Epidemiology Reference Group. Error bars indicate 95th percentile CrIs. A pseudo log transformation is done for values between 0 and 1 in panel B. CHERG, Child Health and Epidemiology Reference Group; CrI, credible interval; GBD, Global Burden of Disease.

(i.e., percent of deaths averted with respect to all neonatal deaths) in Pakistan is smaller compared to other countries considered, we estimate that Pakistan will have one of the greatest absolute reductions in neonatal mortality (i.e., a high number of absolute deaths/cases averted; Fig 1B and 1C), consistent with previous work that Pakistan experiences a significant *K. pneumoniae* burden from nosocomial outbreaks [5].

Across all WHO regions, *K. pneumoniae* isolates are highly resistant to most antibiotics (Fig 2). We focused on those antibiotics often prescribed to treat neonatal sepsis: ampicillin, gentamicin, ceftazidime, amikacin, and meropenem (Fig G in S1 Text for all antibiotics used to treat sepsis) [7,25]. Prevalence of resistance to ampicillin is highest: 89.82% [CrI: 64.97 to 99.49] of *K. pneumoniae* isolates are resistant to ampicillin worldwide (Fig 2B). The average rate of resistance to gentamicin, which is usually prescribed in combination with ampicillin as the first-line treatment for sepsis per current WHO guidelines [32], is 57.22% [CrI: 31.540 to 80.42] (Fig 2B). Comparatively, fewer isolates were resistant to ceftazidime and amikacin—frequently employed as a second-line combination to treat neonatal sepsis—with 44.63% [26.84 to 72.20] and 35.80% [14.69 to 58.53], respectively, isolates identified as resistant (Fig 2B). Approximately 22.43% [5.24 to 41.42] of isolates from neonates who died were resistant to meropenem.

Because ampicillin and gentamicin are prescribed together as the first-line treatment for neonatal sepsis in many countries, we considered the joint distribution of sepsis isolates that were resistant to neither, one, or both antibiotics (Fig 2C and 2D). Resistance is high against both drugs, with an average of 86.38% [CrI: 73.09 to 94.74] of isolates resistant to both drugs across all regions (Fig 2D). Notably, rates of resistance to ampicillin alone were substantially higher than to gentamicin alone (Fig 2D). In fact, less than 5% of isolates were resistant to just gentamicin (Fig 2D). Consequently, resistance to gentamicin almost guaranteed resistance to ampicillin (Fig 2D).

Different antibiotics within the same antibiotic class display varying geographic trends. For instance, resistance to ampicillin is approximately constant across all geographic regions. Comparatively, rates of resistance to ceftazidime and meropenem are variable across geographic regions: Rates of resistance to ceftazidime in Africa are high and close to those of

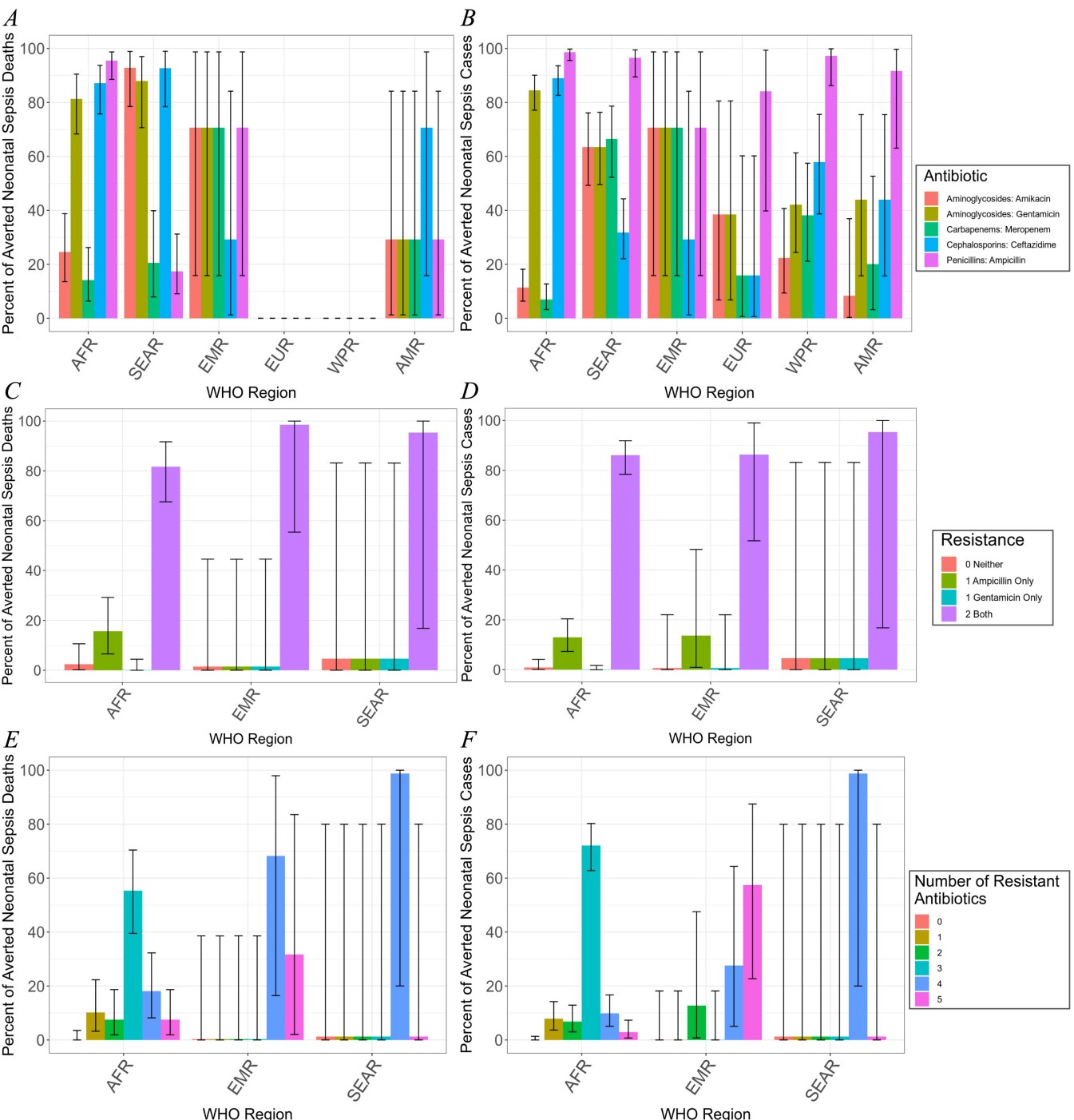

**Fig 2. AMR distribution of vaccine-avertable deaths and cases.** (A, B) Median percent of isolates that are resistant to specific classes of antibiotics by WHO region. Error bars indicate 95th percentile CrIs. (A) Of those deaths that are averted, the percent of deaths that are caused by *K. pneumoniae* resistant to each class of antibiotics. (B) Of those cases that are averted, the percent of illnesses that are caused by *K. pneumoniae* resistant to each class of antibiotics. (C, D) Median percent of isolates that are resistant to multiple classes of antibiotics. Error bars indicate 95th percentile CrIs. (C) Of those deaths that are averted, the percent of isolates that have varying degrees of MDR. (D) Of those cases that are averted, the percent of isolates that have varying degrees of MDR. (E, F) Joint distribution of ampicillin and gentamicin resistance. Error bars indicate 95th percentile CrIs. (E) Of those deaths that are averted, the percent of isolates that are resistant to combinations of aminoglycoside and beta lactam drugs. (F) Of those cases that are averted, the percent of isolates that are resistant to combinations of aminoglycoside and beta lactam drugs. WHO regions with no data indicate that no isolates were gathered in those regions, so no informative modeling can be done. AMR, antimicrobial resistance; CrI, credible interval; MDR, multidrug resistance; WHO, World Health Organization.

ampicillin, approximately 60 percentage points greater than rates of resistance to meropenem. Resistance to meropenem is low (7.01% [CrI: 3.21 to 12.64]) in Africa and substantially higher in South-East Asia (66.58% [CrI: 52.68 to 78.82]), greater than resistance to ceftazidime (Fig 2B). Among aminoglycosides, resistance rates to amikacin and gentamicin are similar in all regions except Africa where gentamicin resistance is 60 percentage points greater than amikacin resistance (Fig 2B). Finally, we consider trends across antibiotics within the same treatment regimen; in particular, we compare ampicillin and gentamicin (first-line treatment) against ceftazidime and amikacin (second-line treatment). While rates of resistance to ampicillin are consistently greater than to gentamicin, rates of resistance to ceftazidime are greater than amikacin only in Africa and the Western Pacific (Fig 2B).

Across all regions, we generally did not find an increase in drug-resistant *K. pneumoniae* isolates from neonates who died compared with all *K. pneumoniae* isolates in general: The prevalence of resistance to a specific drug class is approximately equal whether the isolate was taken from a septic neonate who died or not (Fig 2A and 2B). However, there are 2 key exceptions: In South-East Asia, rates of resistance to amikacin, gentamicin, and ceftazidime are higher in neonates who died, and rates of resistance to ampicillin are lower among all neonates. Nevertheless, we do not find that isolates from neonates who succumbed to sepsis were more likely to exhibit AMR (Fig 2A and 2B), resistance to ampicillin or gentamicin (Fig 2C and 2D), or higher MDR (Fig 2E and 2F) in comparison to all isolates.

Most *K. pneumoniae* have become resistant to multiple antibiotics (Fig 2E and 2F). On average, more than 99% of isolates were resistant to at least 1 drug. Many isolates were resistant to multiple antibiotics, and isolates that were resistant to none were rare (0.15% [CrI: $2.35 \times 10^{-9}$ to 18.17]). The highest MDR rates are observed in the Eastern Mediterranean and South-East Asia where over 98% of *K. pneumoniae* are resistant to either 4 or 5 different antibiotics (Fig 2F); however, in Africa, most isolates are resistant to 3 antibiotics only with fewer isolates resistant to 4 or 5 antibiotics (Fig 2F).

### Projected global and future benefits of a *K. pneumoniae* maternal vaccine

We expanded our estimates of the effects of a maternal *K. pneumoniae* vaccine on childhood mortality and neonatal sepsis to all countries using regression models as described above (Table C in S1 Text, $R^2$ of 84.0%, $p < 0.001$). We project that 14.91% [CrI: 5.26 to 21.24] of all neonatal sepsis deaths in countries of interest (all countries except for high-income nations)—that is, 80,258 [CrI: 18,084 to 189,040] neonatal deaths or 3.40% [CrI: 0.75 to 8.01] of all neonatal mortality—and 399,015 [CrI: 334,523 to 485,442] neonatal sepsis cases could be averted yearly once vaccination coverage reaches that of the maternal tetanus vaccine (Figs 3A and 3B, and Figs H and I in S1 Text for the 95th percentile CrIs, and S2 Table for the numeric values). We project the greatest reduction in overall neonatal mortality (i.e., fraction of deaths reduced due to vaccination) to be in Africa, specifically in sub-Saharan Africa, with a significant but lesser reduction in South-East Asia, the Eastern Mediterranean, and parts of Latin America. The greatest decreases in absolute deaths from neonatal sepsis are in India followed by sub-Saharan Africa; there were approximately equal decreases in the number of deaths in the other regions.

In general, we project regions with the greatest reduction in *K. pneumoniae* neonatal sepsis are likely to experience the greatest reduction in neonatal deaths due to antimicrobial-resistant *K. pneumoniae* sepsis (Figs 3C and 3D and Figs J–L in S1 Text). However, whereas overall neonatal mortality would be considerably reduced across Africa and South Asia, the greatest reduction in resistant *K. pneumoniae* deaths would occur in Central Africa, East Africa, and India (Fig 3). Antibiotic resistance remains a significant issue in the treatment of *K.*

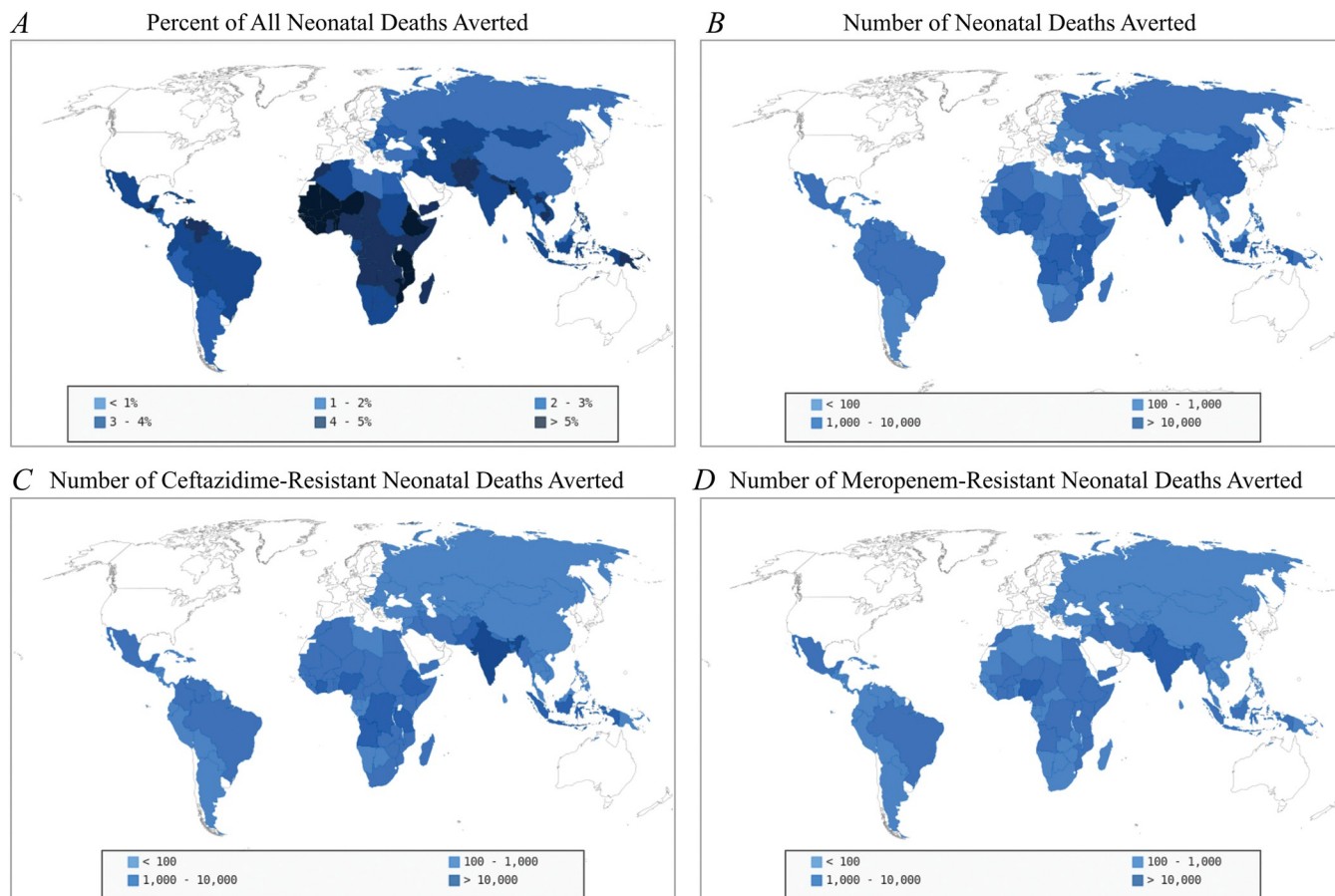

**Fig 3. Projected global impact of a maternal *K. pneumoniae* vaccine.** (A) Projected percent of neonatal deaths with respect to all neonatal deaths that could be averted due to a maternal *K. pneumoniae* vaccine. (B) Projected median number of neonatal sepsis deaths averted due to maternal vaccination. (C) Projected number of ceftazidime-resistant neonatal sepsis deaths averted due to maternal vaccination. (D) Projected number of meropenem-resistant neonatal sepsis deaths averted due to maternal vaccination. The maps are reprinted from pygal_maps_world under GNU GPL.

*pneumoniae* because the projected number of both ceftazidime-resistant deaths (Fig 3C) and meropenem-resistant deaths (Fig 3D) is only marginally less than the projected number of total sepsis deaths (Fig 3B and S2 Table). There are likely to be more ceftazidime-resistant deaths than meropenem-resistant deaths averted globally. However, the relative burden of ceftazidime resistance is greater in North Africa while the burden of meropenem resistance is greater in South Asia. Regardless, for both drugs, we estimate high resistance rates worldwide and a significant number of deaths caused by drug-resistant infections could be averted through maternal vaccination.

Finally, we project the benefits of a maternal vaccine in future scenarios of increased AMR prevalence. Available *K. pneumoniae* genomes indicate that the prevalence of carbapenemase ARGs (CRGs) is increasing significantly worldwide (Fig 4A) at a rate of 0.050 [95th percentile linear probability model confidence interval: 0.042 to 0.058, $p < 0.001$] ARGs being acquired per isolate yearly (Table D in S1 Text; $R^2 = 91.1\%$). Comparatively, aminoglycoside ARGs (AGRGs) increased significantly in the early 2000s (Fig 4A), but subsequently have not increased significantly ($p = 0.436$) and are constant at an average of 2.69 AGRGs per isolate.

Based on the increasing rates of CRGs, we extrapolated the trends from historic data to estimate the number of CRGs present in 2030 and calculated the associated future burden of

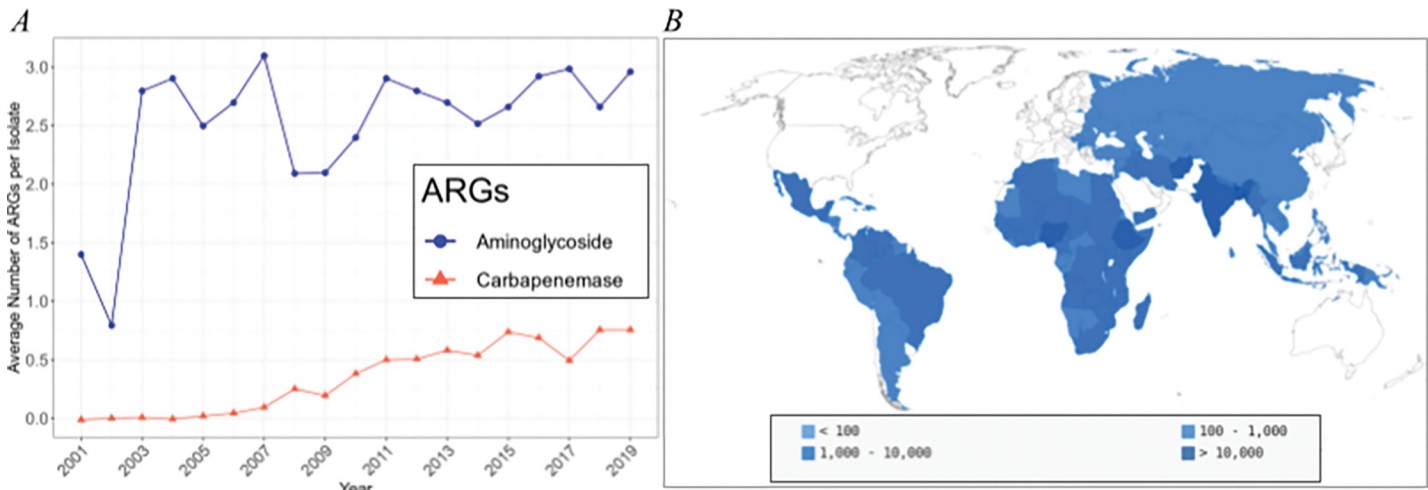

**Fig 4. Projected benefits of a maternal vaccine under future AMR scenarios.** (A) Average number of ARGs that confer resistance to aminoglycosides ($N$ = 24,359) and carbapenems ($N$ = 5,189) per isolate by year of collection of *K. pneumoniae* isolates. (B) Projected median number of meropenem-resistant neonatal deaths averted by 2030 due to maternal vaccination based on the trends shown in Fig 4A. The maps are reprinted from pygal_maps_world under GNU GPL. AMR, antimicrobial resistance; ARG, antimicrobial resistance gene.

meropenem-resistant *K. pneumoniae* deaths that could be averted by a potential maternal vaccine (Figs 4B and Fig M in S1 Text for the 95th percentile CrIs). The map both illustrates neonatal survival trends similar to those described above and the new regional benefits from vaccination as AMR increases. Countries that may experience only marginal benefits from vaccine distribution—such as those in the Americas including Venezuela and Colombia, those in the Western Pacific including Indonesia, those in the Eastern Mediterranean including Afghanistan, and those in South-East Asia including Bangladesh—are likely to experience greater reductions in neonatal mortality when accounting for increases in AMR.

## Discussion

*K. pneumoniae* is the leading cause of newborn sepsis deaths worldwide (Figs O–U in S1 Text); however, no *K. pneumoniae* vaccine is currently approved. We estimate the global benefits of a potential maternal *K. pneumoniae* vaccine in reducing neonatal mortality and AMR. A *K. pneumoniae* vaccine has the potential to significantly improve child wellbeing, reduce a large fraction (15%) of neonatal deaths and sepsis infections, and will benefit additional geographic areas as AMR is projected to increase. Most neonatal sepsis cases are multidrug resistant, further underscoring the benefits of vaccination. Nevertheless, our modeling suggests that the bacteria causing neonatal sepsis in neonates that eventually die do not have higher antimicrobial or MDR rates. Instead, resistance is high in all isolates and against all drugs, underscoring neonatal fragility and suggesting that other factors such as access to effective drugs and antibiotic dosing regiments or standard of care may be driving differential mortality [7]. Regardless, our estimates indicate that maternal vaccination against *K. pneumoniae* would have global benefits and improve efforts toward reaching child survival targets in LMICs worldwide.

Our methodology represents an advantage over using just clinical data or a single study to draw conclusions and make estimates: We applied a Bayesian framework to aggregate data from various studies and generate increased predictive power. This approach more accurately reflects the underlying uncertainty and allows meaningful inference and projections, even when there are few reported and annotated cases of culture-confirmed neonatal sepsis.

However, further work is required to develop a vaccine that is approved, widely distributable, and affordable. Most *K. pneumoniae* neonatal sepsis cases occur in premature infants [26], but the success of a maternal vaccine will depend on robust transplacental antibody transfer from mother to a developing fetus, which peaks late in the third trimester of pregnancy. Additional evaluation of the adequacy of maternal antibody transfer for premature infants is important to inform maternal vaccine development efforts.

There are several potential caveats in our study. Our estimates of averted cases and deaths assume that vaccination rates instantly reach their maximum in all countries—that is, we do not model the reduction in cases of *K. pneumoniae* sepsis as vaccine rollout progresses. To address this, we have conducted sensitivity analyses where we vary the vaccine efficacy/coverage (S2 Table). Our estimates for the number of avertable deaths respond linearly to vaccine efficacy and coverage. Other studies that estimate the effect of a hypothetical intervention against a communicable disease have used a similar assumption [33,34]. As more details of the potential *K. pneumoniae* vaccine are determined, we can refine our model accordingly. Since little is known about the factors that drive nosocomial infections and hospital-acquired neonatal sepsis, the regression models we used to produce estimates of averted cases in all countries are simple and consider only health variables. Nevertheless, previous work that has attempted to extrapolate disease burden and rates of AMR across countries has used a similar econometric-like approach [29]. Finally, our estimates of the CFR of *K. pneumoniae* neonatal sepsis may be biased due to loss-to-follow-up: In particular, we are unable to account for whether there is a sampling bias of a disproportionate number of deaths being recorded. Further data is required to address this issue.

The uncertainty in our estimates largely comes from uncertainty in the underlying values for annual neonatal sepsis deaths from CHERG and GBD, which is considerable. Although estimates for the percentage of averted deaths are consistent (Fig 1A), estimates for the number of avertable deaths using data from CHERG as opposed to GBD often differ by an order of magnitude (Fig 1B). This is because of underlying differences in the number of neonatal sepsis deaths from the 2 sources. Estimates using CHERG data consistently indicate a higher number, though estimates derived from GBD data are still substantial. In many cases, the CrIs from the estimates using different data sources overlap. To address this issue, better country-level data on the burden of neonatal sepsis are needed. Moreover, the sample sizes from all 3 studies were limited and we were unable to delineate hospital outbreaks of *K. pneumoniae* caused by a single strain that may have differential vaccine aversion. We have aimed to address this limitation using the Bayesian approach and noninformative prior, as discussed above, to enable meaningful inference while considering latent noise in the data.

Finally, because of the limited data available on the etiologies causing neonatal sepsis, our analysis only considers country-level trends for *K. pneumoniae* neonatal sepsis cases and WHO regional trends for AMR. However, there is heterogeneity within a country and region; our modeling is unable to capture that. We do not know whether heterogeneity within a country is due to external factors or rather indicates study-level variability: Given our limited data and few countries for which we have multiple study sites, we do not model study-level variability, though this is an area of future work. Additional work is required to better understand within-country and within-region trends: Preliminary data indicates variability in sepsis levels within a country and across private and public hospitals [5,35,36]. Moreover, we may be subject to a sampling bias because the data from the 3 studies we used are point estimates of the impact of sepsis at specific healthcare facilities within a country rather than surveillance over the whole country. In particular, we must limit our analysis to cases of culture-confirmed *K. pneumoniae*, but doing so makes our estimates a probable lower bound on the actual number of averted deaths and cases: it is likely that not all cases of neonatal sepsis are identified as such

and cultured. Even if they are cultured, they may not test positive due to low blood volume, a bias whose impact across study sites likely varies appreciably but is unknown [37]. Additional work is required to better understand the factors that impact neonatal sepsis rates within a region. We also lacked data from many countries. We tried to minimize the effect of this by projecting the number of averted cases by country of maternal vaccination on antimicrobial-resistant *K. pneumoniae* neonatal sepsis on a regional basis and limiting our extrapolations to LMICs. Moreover, we used a uniform prior over the probability of mortality from *K. pneumoniae* to consider the variability in *K. pneumoniae* mortality across all countries in a region: This ensures that the CrIs for our projections to other countries include other mortality rates that may be the true underlying mortality rate for that country. Nonetheless, additional data on *K. pneumoniae* neonatal sepsis across more countries is required to reduce the uncertainty in our estimates.

Future work includes a cost-effectiveness analysis to estimate the economic and social implications of maternal vaccination. A *K. pneumoniae* vaccine would reduce the economic burden of infection for both the patient and the hospital: quantifying these impacts is critical to understanding the greater societal impacts of this vaccine, especially as the burden of sepsis is highest in LMICs that have larger health expenditure (Fig N in S1 Text). Additionally, *K. pneumoniae* is known to be the leading driver of sepsis, especially in the late neonatal stage: reducing sepsis during this timeframe would be significantly advantageous to maternal admissions and mitigating healthcare burden as it would reduce the length of prolonged hospital stays. Beyond reducing cases and deaths, a *K. pneumoniae* vaccine would likely also reduce antibiotic usage and thus may help improve antibiotic stewardship in LMICs and reduce AMR rates. It is noteworthy that in many LMICs, antibiotic availability is challenging (particularly for the more potent drugs) and the cost is in many cases deferred to the family; therefore, a maternal vaccine will also alleviate drug demand and provide local financial benefits. Moreover, vaccination may reduce the severity of disease and have positive social ramifications by reducing stigma surrounding infection and helping mothers who would otherwise leave their jobs to care for sick newborns [38]. We have highlighted the main benefits of the rollout of a potential *K. pneumoniae* maternal vaccine: reducing global cases and deaths of resistance and susceptible neonatal sepsis, reducing overall neonatal mortality, and improving childhood health.

## Supporting information

**S1 Text. Supplementary materials.** Fig A. Raw data. Calculated percent of neonatal sepsis deaths that are associated (i.e., an isolate from the neonate who died was culture-positive) with various etiologies across each study by location. Table A. Number of neonates who died of neonatal sepsis divided by number of neonates surveilled by location. Fig B. Flow diagram summarizing cases of culture-confirmed sepsis used in the main analysis of vaccine-avertable sepsis and AMR. Fig C. Flow diagram summarizing data collection and cleaning of the *Klebsiella pneumoniae* genomes used in the antimicrobial resistance genes (ARG) prevalence analysis. BARNARDS refers to data gathered from the Burden of Antimicrobial Resistance in Neonates in Developing Societies study. Fig D. Distribution of available genomic data from PathogenWatch and the Burden of Antimicrobial Resistance in Neonates from Developing Societies study by year used in the antimicrobial resistance gene prevalence analysis. Fig E. Tree map of the distribution of available *K. pneumoniae* isolates across countries for use in the prevalence of antimicrobial resistance genes analysis. Colors have no meaning and are used to create contrast between countries. Fig F. Schematic diagram of the modeling framework. Table B. Model parameters. Note that this refers to values that are used as inputs to various

modeling stages, not quantities that are predicted through the modeling process described in Fig F. Fig G. Raw resistance data by study. Table C. Regression analysis results for the model used to extrapolate the number of averted deaths from those countries for which we have data to all countries. Fig H. 2.5th percentile (i.e., lower bound) of estimates represented as maps shown in Fig 3. The maps are reprinted from pygal_maps_world under GNU GPL. Fig I. 97.5th percentile (i.e., lower bound) of estimates represented as maps shown in Fig 3. The maps are reprinted from pygal_maps_world under GNU GPL. Fig J. As Fig 3C and 3D but for ampicillin. Median estimates shown on top. 2.5th percentile shown in middle. 97.5th percentile shown on bottom. The maps are reprinted from pygal_maps_world under GNU GPL. Fig K. As Fig 3C and 3D but for gentamicin. Median estimates shown on top. 2.5th percentile shown in middle. 97.5th percentile shown on bottom. The maps are reprinted from pygal_-maps_world under GNU GPL. Fig L. As Fig 3C and 3D but for amikacin. Median estimates shown on top. 2.5th percentile shown in middle. 97.5th percentile shown on bottom. The maps are reprinted from pygal_maps_world under GNU GPL. Table D. Regression analysis results for the model used to estimate the yearly rate of increase in antimicrobial resistance genes. Fig M. Credible interval of map shown in Fig 4B. 2.5th percentile shown on top and 97.5th percentile shown on bottom. The maps are reprinted from pygal_maps_world under GNU GPL. Fig N. Health expenditure as a fraction of the country's GDP. Data courtesy of the WHO, Global Health Observatory (2022). Map reprinted from OurWorldInData under a CC-BY license. Original: https://ourworldindata.org/grapher/total-healthcare-expenditure-gdp. Fig O. As Fig 1A but for other etiologies of interest. Median estimated fraction of neonatal deaths averted given maternal vaccination against a specific pathogen at 70% efficacy and coverage equivalent to that of the maternal tetanus vaccine. Median shown; error bars indicate 95th percentile Bayesian credible intervals. GBD refers to data from the Global Burden of Disease study, and CHERG refers to data from the Child Health and Epidemiology Reference Group. Fig P. As Fig 1B but for other etiologies of interest. Median estimated number of avertable neonatal sepsis deaths given maternal vaccination against a specific pathogen at 70% efficacy and coverage equivalent to that of the maternal tetanus vaccine. Median shown; error bars indicate 95th percentile Bayesian credible intervals. A pseudo log transform is done for values between 0 and 1. GBD refers to data from the Global Burden of Disease study, and CHERG refers to data from the Child Health and Epidemiology Reference Group. Fig Q. As Fig 1C but for other etiologies of interest. Median estimated number of avertable neonatal sepsis cases given maternal vaccination against a specific pathogen at 70% efficacy and coverage equivalent to that of the maternal tetanus vaccine. Median shown; error bars indicate 95th percentile Bayesian credible intervals. GBD refers to data from the Global Burden of Disease study, and CHERG refers to data from the Child Health and Epidemiology Reference Group. Fig R. As Fig 2A but for other etiologies and relevant antibiotics of interest. Estimated median fraction of isolates from neonates who died with culture-confirmed sepsis that are resistant to various drugs across WHO regions. Median shown; error bars indicate 95th percentile Bayesian credible intervals. Fig S. As Fig 2B but for other etiologies and relevant antibiotics of interest. Estimated median fraction of isolates from neonates with culture-confirmed sepsis that are resistant to various drugs across WHO regions. Median shown; error bars indicate 95th percentile Bayesian credible intervals. Fig T. As Fig 2E but for other etiologies of interest. Antibiotics considered are shown in Figs R and S. Median shown; error bars indicate 95th percentile Bayesian credible intervals. Fig U. As Fig 2F but for other etiologies of interest. Antibiotics considered are shown in Figs R and S. Median shown; error bars indicate 95th percentile Bayesian credible intervals.
(PDF)

**S2 Text. Extended methods.** Derivation of the mixture model posterior distribution (Section 1) and details of antimicrobial susceptibility testing done (Section 2).
(PDF)

**S1 Table. Details of the analyzed genomes used in this study including FASTA file ID, year, source, country of origin, and number of identified antimicrobial resistance genes.**
(XLSX)

**S2 Table. Impact of maternal vaccination by country.** Country-specific estimates of overall averted cases, deaths, and result on neonatal mortality as well as antibiotic resistant deaths.
(XLSX)

## Acknowledgments

We thank the BARNARDS and NeoObs groups for sharing their data. In particular, TRW would like to personally thank Kathryn Thomson and Rebecca Milton, both part of the BARNARDS group. We thank Dianna Blau from the CHAMPS group for valuable feedback. We would also like to thank each CHAMPS site team. We thank Nicole Benson, at the Bill & Melinda Gates Foundation, Mateusz Hasso-Agopsowicz and Isabel Frost, both at the World Health Organization, and Sonya Davey, at Brigham and Women's Hospital for valuable feedback. We thank Sally Atwater for valuable editorial suggestions on drafts.

## Author Contributions

**Conceptualization:** Timothy R. Walsh, Padmini Srikantiah, Ramanan Laxminarayan.

**Data curation:** Kirsty Sands, Timothy R. Walsh, Ramanan Laxminarayan.

**Formal analysis:** Chirag K. Kumar, Kirsty Sands.

**Funding acquisition:** Ramanan Laxminarayan.

**Supervision:** Timothy R. Walsh, Ramanan Laxminarayan.

**Visualization:** Chirag K. Kumar.

**Writing – original draft:** Chirag K. Kumar.

**Writing – review & editing:** Chirag K. Kumar, Kirsty Sands, Timothy R. Walsh, Seamus O'Brien, Mike Sharland, Joseph A. Lewnard, Hao Hu, Padmini Srikantiah, Ramanan Laxminarayan.

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
