## [Editor Report · Decision Letter 0]

14 Oct 2022

Dear Dr Laxminarayan, 

Thank you for submitting your manuscript entitled "Global, regional, and national estimates of the impact of a maternal Klebsiella pneumoniae vaccine: A Bayesian modeling analysis" for consideration by PLOS Medicine.

Your manuscript has now been evaluated by the PLOS Medicine editorial staff as well as by an academic editor with relevant expertise and I am writing to let you know that we would like to send your submission out for external peer review.

Please re-submit your manuscript within two working days, i.e. by Oct 18 2022 11:59PM.

Kind regards,

Philippa Dodd, MBBS MRCP PhD

Editor

PLOS Medicine

---

## [Decision Letter · Decision Letter 1]

26 Jan 2023

Dear Dr. Laxminarayan,

Thank you very much for submitting your manuscript "Global, regional, and national estimates of the impact of a maternal Klebsiella pneumoniae vaccine: A Bayesian modeling analysis" (PMEDICINE-D-22-03384R1) for consideration at PLOS Medicine. 

[LINK]

In light of these reviews, I am afraid that we will not be able to accept the manuscript for publication in the journal in its current form, but we would like to consider a revised version that addresses the reviewers' and editors' comments. Obviously we cannot make any decision about publication until we have seen the revised manuscript and your response, and we plan to seek re-review by one or more of the reviewers. 

We expect to receive your revised manuscript by Feb 16 2023 11:59PM. Please email us (plosmedicine@plos.org) if you have any questions or concerns.

We look forward to receiving your revised manuscript. 

Sincerely,

Philippa Dodd, MBBS MRCP PhD

PLOS Medicine

plosmedicine.org

GENERAL

Please respond to all editor and reviewer requests detailed below, in full.

Please include line numbers starting at 1 on the title page and in continuous sequence thereafter

COMPETING INTERESTS STATEMENT

“The sponsor of the study provided input in the writing of the report, acting in a purely scientific capacity.” Please elaborate further on the precise role that the funder played in writing the manuscript. What does “purely scientific capacity” encompass? Please include details of each individual affiliated to the funding body that contributed to manuscript and provide details of each individual’s specific role in the preparation of the manuscript.

DATA AVAILABILITY STATEMENT

Thank you for agreeing to make your data available upon request. The Data Availability Statement (DAS) requires revision. For each data source used in your study: 

RESEARCH IN CONTEXT

Please remove this section and all sub-sections from the manuscript

ABSTRACT

Please structure your abstract using the PLOS Medicine headings (Background, Methods and Findings, Conclusions).

Please combine the Methods and Findings sections into one section, “Methods and findings”.

Abstract Background: 

Please ensure that the final sentence clearly states the study question. 

“…resistant to antibiotic treatment regimens…” suggest introducing carbapenem resistance here at the outset, rather than later

Please define “K. pneumoniae” after the full name in the lines above, prior to first use here

Abstract Methods and Findings:

“We used data from three global studies…” It would be helpful to include further details here - which countries did those studies include, how many participants, over what time frame (s) how many had K. pneumoniae and so on

“We quantified the rate of acquisition of antimicrobial resistance genes in K. pneumoniae isolates…” it would be helpful to include details of how you did this

“…predict the future burden…” how do you define “burden”?

Please quantify the main results with p values, as well as 95% CrIs. When reporting p values please report as p<0.001 and where higher the exact p-value as p=0.002, for example. If not including p-values, not for the purpose of transparent data reporting please provide reasons as to why not. 

Please ensure that all numbers presented in the abstract are present and identical to numbers presented in the main manuscript text.

Please provide the actual numbers of events for the outcomes, not just summary statistics/percentages 

Did you adjust your analyses for any variable factors? If so, please also detail these

In the last sentence of the Abstract Methods and Findings section, please describe the main limitation(s) of the study's methodology.

AUTHOR SUMMARY

METHODS and RESULTS

We ask the following of all authors of modelling studies. Please see below and in your rebuttal, please sign post to where in the manuscript the relevant information can be located:

• Please provide a diagram that shows the model structure, including how the disease natural history is represented, the process and determinants of disease acquisition, and how the putative intervention could affect the system.

• Please provide a complete list of model parameters, including clear and precise descriptions of [the meaning of each parameter, together with the values or ranges for each, with justification or the primary source cited, and important caveats about the use of these values noted].

• Please provide a clear statement about how the model was fitted to the data [including goodness-of-fit measure, the numerical algorithm used, which parameter varied, constraints imposed on parameter values, and starting conditions].

• For uncertainty analyses, please state the sources of uncertainties quantified and not quantified [can include parameter, data, and model structure].

• Please provide sensitivity analyses to identify which parameter values are most important in the model. Uncertainty estimates seek to derive a range of credible results on the basis of an exploration of the range of reasonable parameter values. The choice of method should be presented and justified.

• Please discuss the scientific rationale for this choice of model structure and identify points where this choice could influence conclusions drawn. Please also describe the strength of the scientific basis underlying the key model assumptions.

We agree with the reviewer (please see below) that additional information regarding the datasets used would be helpful. Please revise accordingly, as for the abstract.

Please remove role of the funding source from the end of the methods section and include only in the manuscript submission form

Please ensure that results are quantified with p-values as well as 95% CrIs. When reporting p-values please report as p<0.001 and where higher, the exact p-value as p=0.002, for example. If not including p-values, not for the purpose of transparent data reporting please provide reasons as to why. 

Please refrain from discussing your results in the results section (see page 11, second paragraph) and report only your findings. Please move any discussion of your results to the appropriate section of your discussion

Please ensure all abbreviations are clearly defined at first use – ARGs and CRGs, for example on page 13 – I couldn’t see a previous definition but my apologies if I missed it

FIGURES

All the figures are very small and therefore not easily interpretable by the reader. Please see below for specific comments. Suggest considering placing a title above each image in the figures to help the reader but we leave this to your discretion.

Please ensure that each figure is affiliated to an appropriate caption which clearly describes its content without the need to refer to the text. Please ensure that any and all abbreviations are defined within the appropriate legend/caption

Please consider avoiding the use of green or red to make figures more accessible to those with colour blindness

Figure 1: we agree with the reviewers that the figure is not very accessible, text is too small and the presentation of different axes is a bit confusing. Please show the axis beginning at zero. If this is not possible, please show a break in the axis.

In the figure caption please clearly state the meaning of the bars and whiskers and please define GBD and CHERG

Figure 2: all the text in the figures in too small for the reader, including the axis labels. The error bars are very distracting and too large, they should not meet each other. The disparity in bar sizes makes it very difficult to read the data presented here

Figure 3: As above – these are also very small and as such it is difficult to appreciate the different colour densities which we like. It may also help the reader to title each of the individual maps within the figure. Please revise accordingly. Please confirm that the appropriate usage rights apply to the use of the maps. Please see our guidelines for map images: https://journals.plos.org/plosmedicine/s/figures#loc-maps

Figure 4: as above, please also define abbreviations - AMR, ARGs, CRGs. The legend is hard to read and could be overlayed on the graph and enlarges, for example.

SUPPORTING FILES

Please ensure that all figures/tables are affiliated to an appropriate caption which clearly describes their contents. Suggest considering applying some universal formatting to the figures throughout the main manuscript and the supporting files. A number of bar graphs are presented but each are very different, please revise accordingly

Kp Appendix – please ensure abbreviations are defined throughout

Fig S1 – please ensure colours are accessible to those with colour blindness

Fig S3 – please define abbreviations

Fig S5 – is difficult to interpret without a legend for the colours, please include

Fig S7 onwards – captions for the maps insufficiently describes what they show. What is meant by lower and upper bound, for example – you can work it out, but it’s not obvious to the reader. Do these bounds need to presented as maps? Perhaps a table would be preferable here?

What are the items listed as 1-9 above where you refer to a CSV file for the “complete table”. Please clarify

Fig S13-S16 - please define the meaning of the bars and whiskers and define the abbreviations GBD and CHERG

Fig S17-20 – see earlier re: size of error bars, colours, uniformity, definitions of all abbreviations

** We note that there are a lot of figures presented here and (as above) we suggest that much of this data would be more accessible if presented as tables **

DISCUSSION

Please remove the declaration of interests and data sharing statements form the end of the discussion – these should be included in the relevant parts of the manuscripts only – they will be complied as meta-data

REFERENCES

Please ensure that your referencing follows PLOS Medicine’s style. Further details can be found here: https://journals.plos.org/plosmedicine/s/submission-guidelines#loc-references

In the bibliography please ensure that you list up to more but no more than 6 author names followed by et al where more than 6 authors contribute. Please ensure that journal name abbreviations are those found in the National Center for Biotechnology Information (NCBI) databases. 

Comments from the reviewers:

Reviewer #1: Overall comment: In general, this is a useful manuscript that brings into focus an important issue, ie how impactful a vaccine against K pneumoniae might be for preventing neonatal sepsis. One major issue I have is that the data sources and assumptions and initial values that go into the model are not described well, which gives the impression that the results are generated from a "black box." I suggest adding a "Table 1" that clearly spells out model parameters. Less importantly, there are some style issue with the manuscript that I suggest cleaning up, in particular the inclusion of methods and discussion points in the results section; including discussion points in the results section is problematic, as it blurs the line between what the model is directly showing and authors' opinions or conclusions. 

Abstract: Fix the introduction paragraph to clarify that the focus is on neonatal sepsis, not sepsis generally, and that data sources were primarily from LMIC.

Page 6, data section: CHAMPS is Child Health and Mortality Prevention Surveillance (ie, not Childhood…). Also, the authors should include a sentence or two about each data source so that readers do not need to hunt to find basic information (e.g., countries involved, who was included, method of enrollment, years, etc). 

Methods, general comment: One tricky issue with maternal immunization as a method to prevent neonatal infections is that babies born preterm are often more susceptible to sepsis and may need to stay in hospital for medical support until they are large enough to go home, which results in exposures to hospital-associated pathogens, including K pneumoniae. Conversely, optimal transfer of antibodies from mom to baby across the placenta occurs late in pregnancy (3rd trimester). Did you account for this issue in the model? I was curious why you selected 70% efficacy, and whether this figure accounted for less benefit in preterm births or imperfect coverage of serotypes or just assumed that efficacy of the vaccine against sepsis caused by vaccine serotypes would be about 70% in term infants. I would include this information in the methods section. 

Results, bottom of page 11. Here the authors have included text that should be in the discussion section, as it seems to include conclusions drawn from the data. For example, this is the wording:

" Nevertheless, we do not find overall trends in isolates from neonates who succumbed to sepsis

having higher rates of AMR (Fig. 2A, 2B), resistance to ampicillin or gentamicin (Fig. 2C, 2D)

or higher multi-drug resistance (MDR) (Fig. 2E, 2F), indicating that antibiotic resistance of the

disease-causing K. pneumoniae alone may not significantly impact neonatal mortality. Instead,

resistance is high in all isolates, against all drugs, suggesting that other factors such as access to

effective drugs and antibiotic dosing regiments may be driving differential mortality." 

I'm trying to understand how the authors came to this conclusion, as the isolates that were collected from living babies with sepsis may have indeed caused their death (unclear if the study followed them until the end of their illness) or perhaps there was some other issue (e.g. contamination). Also, since the isolates from deceased children and living children came from two different studies with different patient populations and treatment settings, I would encourage caution in this interpretation. Also, what antibiotics each child received is likely unknown.

Page 12, 5th line under "Projected Global…" heading: typo with the comma placement in the last number in this line.

Page 12, 3rd line from the bottom: How confident do you feel about extrapolating to North America and Europe from the data sets that included little to no data from high income settings? Should those settings be excluded, or more data sources added to better support numbers from high income settings?

Page 16, discussion: In the limitations paragraph, the authors mention that the data included did reflect entire countries, but the data sources only included data from a limited number of countries overall. Isn't the complete lack of data from many countries also an issue that should be mentioned? For example, were Europe and North America extrapolated from Greece and Brazil, respectively?

Figures, general comment: The pictures are quite small relative to the text, so I find myself having to zoom in to interpret the figure then zoom out to read the information about the figure. Perhaps the journal has a way to fix this mismatch, or maybe the labeling on the figures could be larger. 

Figure 1. Is there something problematic with the way the model is handling number of deaths for Greece? The B figure suggests the vaccine is causing deaths, although with the log scale and no zero it's difficult to tell, and the figure with percentages suggests a small percent of deaths are prevented. 

Reviewer #2: SUMMARY

This manuscript reports estimates of the global number of neonatal deaths due to Klebsiella pneumoniae, and the number of these deaths that could be averted by a vaccine with 70% efficacy. In general the approach seems reasonable, though more detail is required on some sections, particularly on how the location-specific estimates were extrapolated to generate regional and global estimates. Specific comments are provided below.

MAJOR ISSUES

* Pages 6-7: the analytic approach appears reasonable if one assumes the fraction of sepsis deaths is homogeneous within the sampled locations, or if the data represent a representative sample of sepsis deaths within each location. It would be useful to examine these assumptions - in particular, where a particular location has several studies, how consistent are these results? If inconsistent, might one explicitly model study-level variation (would lead to greater uncertainty in estimated results, which would be appropriate if there were substantial study-level variation). I think ideally study-level variation would be examined and modelled (if there are enough locations with multiple studies), otherwise this could also be discussed in the limitations. 

* Page 7: for locations with several studies, it is not clear to me why the data were not pooled in a single beta distribution. The approaches will give different answers, though perhaps trivially so, as the current approach adds a value of 1.0 to both beta parameters for each study, via the beta(1,1) prior. I can see the benefit of estimating studies results separately if there was a desire to model study-level variation (I think a beta-binomial model could do this?), yet this is not done (or at least is not described), per comment above. 

* Page 8: the regression approach used to extrapolate to other countries is not described in sufficient detail. Given how important this step is to the analysis (going from 18 countries to global), it would be useful to provide additional information on covariates, functional form, evidence of model fit, etc.

MINOR ISSUES

* Page 2, abstract: If the word count allows, it would be useful to provide more specific details on the scenarios that produced the vaccine impact estimates (e.g., is this vaccination of all pregnant women globally, or some subset?)

* Page 2, abstract: is it possible to provide some measure of uncertainty around the results presented?

* Page 2, abstract: small point, but it is unclear to me what "The most significant benefits" means. Is this to be read as "largest relative benefit"? Perhaps more specific language would be useful.

* Page 6: it would be useful to expand the section describing the data, in particular to include the number of samples and distribution across regions. 

* Page 6 (data): please provide some comment on possible bias that comes with restricting analyses to culture-confirmed cases (and what fraction of cases this generally represents). I realize that using culture-confirmed cases may be the only feasible approach, but still useful to know if there are any biases could result, and how they are dealt with. 

* Page 7: around "effective coverage level equal to that of the maternal tetanus vaccine", it would be useful to give the median and IQR (or something similar) to give the reader some idea of the average level and variation in vaccine coverage assumed for the analysis. 

* Page 7: I might describe what you used the LHS sample for, for readers who are unfamiliar. 

* Page 10: "The average rate of resistance to gentamicin, which is usually prescribed in combination with ampicillin as the first- line treatment for sepsis per current WHO guidelines,32 is 57.22% [CrI: 31.540-80.42]" - I am a bit surprised by how broad the intervals are. Is this because of a small number of observations (i.e. individuals), or the extrapolation process, or something else? On a similar subject, there is substantial uncertainty around Panel A in Figure 1, but not Panels B and C. I realize that B and C are log-scale, but it appears Panel A still has more relative uncertainty (take Thailand for example where in Panel A the interval ranges from ~0 to >3 times the point estimate). Is everything here correct? 

Reviewer #3: According to the authors, there has been no study projecting the effects of K. pneumoniae vaccine distribution. Given that there are numerous efforts in developing a viable K. pneumoniae vaccine, the authors tried to estimate the global, regional, and national impact of a hypothetical vaccine with 70% efficacy on neonatal sepsis infections and mortality using a Bayesian mixture-modeling framework. The estimated impact was presented in terms of number of neonatal sepsis deaths and neonatal sepsis cases averted due to the hypothetical vaccine. In addition, the authors also examined the AMR of sepsis-causing isolates and tried to project the future benefits of maternal K. pneumoniae vaccination. Overall, this is a well-designed and conducted study and the results of the study are of relevance, particularly for LMICs in Africa. Below are my specific comments. 

1. Please add line numbers for easier reference. 

2. The sponsor of the study provided input in the writing of the report, which might result in conflict of interest. 

3. Data used in the study are not fully available. 

4. Page 7, method: "We extended this analysis to estimate the absolute number.. we report values derived using CHERG data." It seems that sometimes you used CHERG data and sometimes you used GBD data to derives the parameter values. However, are data from CHERG and from GBD comparable to each other? Did you do any adjustment to the GBD data to make them comparable to the CHERG data? I did not see you mention this in the method section. 

5. Page 7, method: "We attempted to minimize the impact of potential.. to estimate the CFR". I don't understand how Bayesian method can help minimize the bias here. Loss to follow-ups are usually informative. I am not sure whether Bayesian method can help reduce the bias due to informative LTFUs. 

6. Method: To estimate the global, regional, and national number of cases and deaths, why did you not use some hierarchical modeling approach? Bayesian framework should be perfect to incorporate hierarchical data structure, which can make the estimates between different levels internally consistent. However, I did not see the hierarchical structure in the method section.

[LINK]

---

## [Decision Letter · Decision Letter 2]

17 Mar 2023

Dear Dr. Laxminarayan,

Thank you very much for re-submitting your manuscript "Global, regional, and national estimates of the impact of a maternal Klebsiella pneumoniae vaccine: A Bayesian modeling analysis" (PMEDICINE-D-22-03384R2) for review by PLOS Medicine.

I have discussed the paper with my colleagues and the academic editor and it was also seen again by 3 reviewers. I am pleased to say that provided the remaining editorial and production issues are dealt with we are planning to accept the paper for publication in the journal.

[LINK]

We look forward to receiving the revised manuscript by Mar 24 2023 11:59PM.   

Sincerely,

Philippa Dodd, MBBS MRCP PhD

PLOS Medicine

plosmedicine.org

Requests from Editors:

GENERAL

Thank you for your very detailed and considered responses to previous editor and reviewer requests. Please see below for further comments/suggestions that we request you address in full.

ABSTRACT

Line 17 – please replace the sub-heading “summary” with “Abstract”

AUTHOR SUMMARY

Thank you for including an author summary some suggestions are detailed below

Line 69: suggest “pre-clinical model” perhaps instead of “animal”

Line 74: To improve accessibility to the reader, suggest the following - “Using a Bayesian mixture-model based on clinical surveillance of neonatal sepsis, we present country-specific estimates for the number of deaths and cases of antimicrobial resistant neonatal sepsis, caused by K. pneumoniae, that would be averted if a vaccine with 70% efficacy was to be given to pregnant mothers.”

Line 82: suggest moving this point to precede that at line 79

Line 85: suggest rewording this statement for improved clarity – “…countries of interest…” is rather vague, perhaps “especially in LMICs” instead, perhaps? And, “…due to all bacteria…” this is a little confusing/vague – do you mean bacteria other than K pneumoniae or all strains of K pneumoniae?

Line 88: Suggest “This would help to achieve targets set by […] for improved global child health and to mitigate against inequity in neonatal survival in low- and middle-income settings compared to high-income settings. Suggest that the blank space in brackets is completed (WHO targets?) 

Line 90: Suggest “Reducing cases of neonatal sepsis by vaccination could also contribute to reduced antibiotic use, subsequent improvements in antimicrobial resistance rates and a reduction in healthcare utilization and expenditure.” Or something similar

METHODS and RESULTS

Please see statistical reviewer (reviewer #3) comments below

COMPETING INTERESTS STATEMENT

Line 492: Thank you for updating your statement. Please remove from the end of the discussion in the main manuscript and include only in the manuscript submission form when you re-submit your manuscript.

FIGURES

Figure 4: in the caption you refer to ARGs (and define them) as well as CRGs (and define these) but CRGs are not detailed anywhere in the legend of figure 4 that is available to me. Please revise as necessary.

SOCIAL MEDIA

To help us extend the reach of your research, if not already done so, please provide any Twitter handle(s) that would be appropriate to tag, including your own, your coauthors’, your institution, funder, or lab. Please detail any handles you wish to be included when we tweet this paper, in the manuscript submission form when you re-submit the manuscript.

Comments from Reviewers:

Reviewer #1: The authors appear to have addressed my concerns. I would ask for one simple correction -- CHAMPS does not do autopsies but rather Minimally Invasive Tissue Sampling (MITS). 

Reviewer #2: Thank you, these revisions resolve my concerns. 

Reviewer #3: I appreciate the authors' responses to my comments. I am happy with most of the responses but I cannot agree with their response to my comment #5. I don't think using a uniform prior can help reduce bias due to lost-to-follow up or any bias.. A uniform prior just means that the analyst knows little/nothing about the quantity of interest and rely almost entirely on the data to inform the estimate. It dose not help with reducing bias at all, and increasing the uncertainty of the estimate does not mean reducing bias either. It just means that you are more uncertain about the estimate.

[LINK]

---

## [Decision Letter · Decision Letter 3]

26 Apr 2023

Dear Dr Laxminarayan, 

On behalf of my colleagues and the Academic Editor, Dr. Rebecca Freeman-Grais, I am pleased to inform you that we have agreed to publish your manuscript "Global, regional, and national estimates of the impact of a maternal Klebsiella pneumoniae vaccine: A Bayesian modeling analysis" (PMEDICINE-D-22-03384R3) in PLOS Medicine.

PRESS

Best wishes,

Pippa 

Philippa Dodd, MBBS MRCP PhD 

PLOS Medicine